

# Morphometric comparisons of plant-mimetic juvenile fish associated with plant debris observed in the coastal subtropical waters around Kuchierabu-jima Island, southern Japan

Alexya Cunha de Queiroz[1], Yoichi Sakai[2], Marcelo Vallinoto[1,3] and Breno Barros[1,2,4]

[1] Instituto de Estudos Costeiros, Laboratório de Evolução, Universidade Federal do Pará, Bragança, Pará, Brazil
[2] Graduate School of Biosphere Science, Laboratory of Aquatic Resources, Hiroshima University, Higashi-Hiroshima, Japan
[3] Centro de Investigação em Biodiversidade e Recursos Genéticos, Universidade do Porto, Vairão, Portugal
[4] Campus de Capanema, Universidade Federal Rural da Amazônia, Capanema, Pará, Brazil

Corresponding author
Breno Barros,
breno_eduardo@terra.com.br

## ABSTRACT

The general morphological shape of plant-resembling fish and plant parts were compared using a geometric morphometrics approach. Three plant-mimetic fish species, *Lobotes surinamensis* (Lobotidae), *Platax orbicularis* (Ephippidae) and *Canthidermis maculata* (Balistidae), were compared during their early developmental stages with accompanying plant debris (i.e., leaves of several taxa) in the coastal subtropical waters around Kuchierabu-jima Island, closely facing the Kuroshio Current. The degree of similarity shared between the plant parts and co-occurring fish species was quantified, however fish remained morphologically distinct from their plant models. Such similarities were corroborated by analysis of covariance and linear discriminant analysis, in which relative body areas of fish were strongly related to plant models. Our results strengthen the paradigm that morphological clues can lead to ecological evidence to allow predictions of behavioural and habitat choice by mimetic fish, according to the degree of similarity shared with their respective models. The resemblance to plant parts detected in the three fish species may provide fitness advantages via convergent evolutionary effects.

## INTRODUCTION

Mimesis is defined as a phenotype evolved in response to selective pressures favouring individuals that can disguise their identity by masquerading as another organism (*Pasteur*, *1982*; *Skelhorn, Rowland & Ruxton*, *2010*; *Skelhorn et al.*, *2010*). Mimesis in fish is a relatively well-studied subject (*Wickler*, *1968*; *Moland, Eagle & Jones*, *2005*; *Robertson*, *2013*), particularly regarding deceptive resemblance to plant parts via protective camouflage, which is a known feature in several freshwater and marine fish species, as extreme crypsis

examples of protective resemblance (*Breder*, *1946*; *Randall & Randall*, *1960*; *Randall*, *2005a*; *Vane-Wright*, *1980*; *Sazima et al.*, *2006*). Although these reports have addressed the patterns and general similarities in morphology or colouration of model plant parts and mimetic fish, few studies have examined similarities among them based on morphological and/or ethological details (*Barros et al.*, *2008*; *Barros et al., 2011*; *Barros et al., 2012*).

Studies focusing on morphology and geometric morphometrics frequently used fish species as models, and several authors have suggested that morphological clues can be used as ecological predictors from basic behavioural constraints, such as swimming mode (*Walker*, *2004*; *Comabella, Hurtado & García-Galano*, *2010*; *Xiong & Lauder*, *2014*), feeding behaviour (*Galis*, *1990*; *Franssen, Goodchild & Shepard*, *2015*) and habitat choice (*Loy et al.*, *1998*; *Gibran*, *2010*; *Soares, Ruffeil & Montag*, *2013*), especially in juvenile fish, suggesting that such changes are important for improving fitness and increasing the chance for survival during subsequent ontogenetic stages (*Barros et al.*, *2011*; *Comabella et al.*, *2013*). Nevertheless, such a tool has not been used to establish comparisons among distant taxa belonging to completely different groups (i.e., fish and plants). In the present study, previously well-known plant-mimetic juvenile fish, the tripletail, *Lobotes surinamensis* (Bloch, 1790), the orbicular batfish, *Platax orbicularis* (Forsskål, 1775) and the ocean triggerfish, *Canthidermis maculata* (Bloch, 1786) were compared with their respective plant models co-occurring in the field to objectively evaluate their resemblance in shape to their respective models. All fish and plant models were observed and sampled from Kuchierabu-jima Island and its surrounding waters, which are subjected to a strong influence of Kuroshio Current.

*Lobotes surinamensis* is generally found in shallow brackish water habitats but may occur far offshore with drifting algae or flotsam, and juveniles may lie on their side matching the colour of the plant debris, from near black to yellow (*Randall*, *2005b*). Juveniles are usually dark-coloured, presenting drifting swimming patterns among dry leaves, exhibiting similar movements to their associated plant model (*Uchida*, *1951*; *Randall*, *2005b*). *Uchida* (*1951*) also described that young *C. maculata* resemble pieces of pine bark and were observed drifting among pieces of bark in a horizontal swimming posture, suggesting mimetic effects. Juveniles of *P. orbicularis* look similar to yellow waterlogged jack tree leaves (genus *Rhizophora*) and greatly resemble floating dead leaves (*Willey*, *1904*; *Breder*, *1946*). *Randall & Randall* (*1960*) reported that larger individuals (87 mm standard length (SL)) resemble large sea hibiscus leaves (*Hibiscus tiliaceus*) with a yellowish-brown colouration, with dorsal and anal fins appearing to lengthen with growth. Such drastic changes in morphological shape occur in juvenile *P. orbicularis* while they maintain a resemblance to drifting leaves (*Barros et al.*, *2015*).

The novel comparative methods presented herein may provide useful associations between behavioural ecology and morphological studies. We tested the null hypothesis of a lack of shape similarity among the studied fish and plant parts, considering both classic and geometric morphometrics comparative approaches. We briefly discuss the functional contributions of camouflage characteristics to fish fitness using mimetic shape attributes as a disguise based on morphological resemblance data among fish and model plants, adopting the concepts of cryptic mimesis as synonym of protective camouflage or masquerading, following the definitions as proposed by *Pasteur* (*1982*), where all fish samples are defined as

"mimetic fish" and all plant part samples as "models," instead of adopting the terminology as proposed by *Skelhorn, Rowland & Ruxton* (*2010*). This is due to the highly dynamic environments such fish usually occur, where mimetic behaviour is achieved not only by appearance, but also through actively behaving alike the drifting models (*Barros et al.*, *2008*; Video S1).

## MATERIAL AND METHODS

### Sampling

Sampling was mainly conducted in the port of Honmura, Kuchierabu-jima Island (Ohsumi Group, 30°28′N, 130°10′E), southern Japan, during diurnal observations July 3–14, 2011 (Fig. S1). The island closely faces the Kuroshio Current and maintains a rich subtropical fish fauna (*Gushima & Murakami*, *1976*). Fish samples and plant debris were collected using hand nets, and the sampled fish were euthanized using 5 ml 95% eugenol in 1 L ethanol as a stock solution. Of this, 20 ml was added to each 1 L of water containing the fish to be euthanized to minimise suffering, following international ethical standards (*Jenkins et al.*, *2014*). All fish samples were preserved in order to maintain integrity of peripheral structures and general shape, and were photographed as soon as possible, in order to avoid any arching or deformation effect from the fixation protocols established (*Valentin et al.*, *2008*). All plant materials were sampled at the island along with their associated fish. As there is no national Japanese licensing framework, samples were collected following the "Guidelines for Proper Conduct of Animal Experiments" set out by the Hiroshima University Animal Research Committee, which are based on international ethical standards, and only after obtaining local community permission.

Fish samples from Kuchireabu-jima Island were identified to as low a taxonomic category as possible, according to available literature (*Nakabo*, *2002*; *Nelson*, *2006*; *Okiyama*, *2014*). Fifteen mimetic fish specimens of three species (Figs. 1A–1C) were observed to drift around plant debris: *Lobotes surinamensis* (Lobotidae; $n = 6$, TL = 3.89 ± 0.46 cm; AVE ± SDEV values), *Platax orbicularis* (Ephippidae; $n = 7$ TL = 2.05 ± 0.42 cm) and *Canthidermis maculata* (Balistidae; $n = 2$, TL = 3.15 ± 0.98 cm). An additional 24 fish specimens ($n = 14$ for *L. surinamensis*, $n = 10$ for *C. maculata*) sampled in subtropical waters of Kagoshima Prefecture were also obtained from the collections of the Kagoshima University Museum (KAUM) to enhance and equalize sample size of our data set for the statistical analyses (see below). The KAUM samples were all juveniles, with relatively similar standard length as those observed (*L. surinamensis* TL = 4.92 ± 2.02 cm, and *C. maculata* TL = 3.95 ± 0.98 cm) and collected near to the present study area, i.e., Satsuma Peninsula of mainland Kagoshima, Tanega-shima Island, and Yaku-shima Island (31°28′–31°33′N, 130°11′–130°51′E) (for details refer to Dataset S1). Of these, the most images were provided by the KAUM ($N = 5$ for *C. maculata* and $N = 11$ for *L. surinamensis*), taken from fresh specimens. All other samples were photographed in the Laboratory of Biology of Aquatic Resources, at the Hiroshima University, and only those with all peripheric structures intact were considered in the analysis. No arched or deformed specimens were used during the analyses, in order to prevent from any misinterpretation of data, as inconclusive attempts

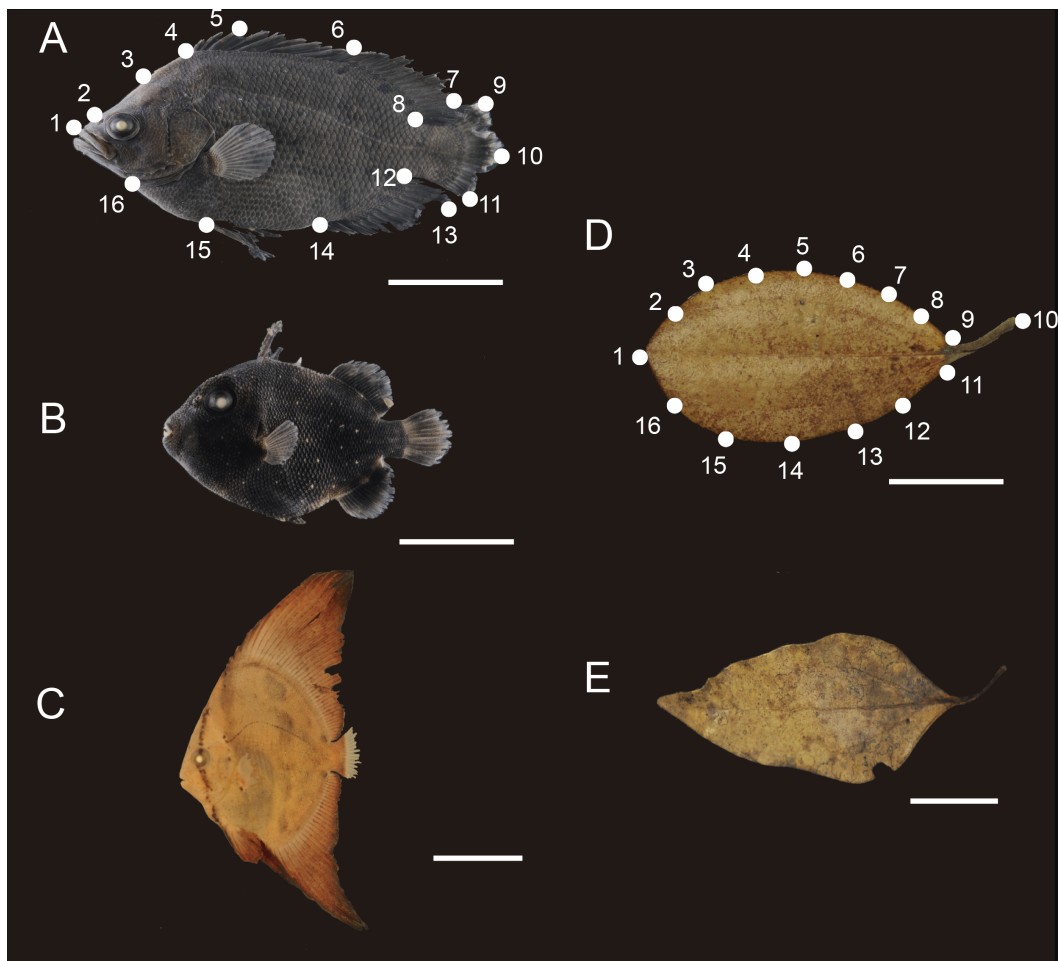

**Figure 1 Mimetic fish and plant models.** Examples of mimetic fish and their models (i.e., floating plant debris) occurring in the shallow waters of Honmura Port, Kuchierabu-jima Island, southern Japan. (A) *Lobotes surinamensis*, (B) *Canthidermis maculata* and (C) *Platax orbicularis* are the mimetic fish observed. The models were subdivided using three criteria of: (D) round leaves, and (E) elongated leaves. The established landmarks and semilandmarks are denoted in (A) for the mimetic fish and in (F) for the models, respectively. White bars indicate 1 cm.

to explain such posture variations by any possible biological factors, as allometric growth or even sexual dimorphism (*Valentin et al.*, *2008*).

Also, additional twelve samples of *P. orbicularis* (TL = 2.05 ± 0.91 cm) collected during previous surveys on Kuchierabu-jima Island (*Barros et al.*, *2008*; *Barros et al.*, *2011*) were eventually employed, in order to equalize *N* size. These were also fixed using the same protocol as standardized herein, being photographed soon after sampling. A total 52 individual mimetic fishes were analysed.

Floating plant debris (hereafter, models, *n* = 43) were collected using hand nets and sorted, then visually subdivided using two subjective criteria (round shapes, as for the Podocarpaceae *Nageia nagi* and the Sapindaceae *Acer morifolium*; or elongated shapes, as for the Laureaceae *Neolitsea sericea* and for the Fagaceae *Castanopsis sieboldii*; Figs. 1D–1E),

**Table 1  List of landmarks.** List of homologous landmarks and criteria adopted for selecting each landmark used for the mimetic fish.

| Landmark | Landmark description |
| --- | --- |
| 1 | Tip of the snout |
| 2 | Nasal cavity |
| 3 | Posterior limit of supra-occipital |
| 4 | Anterior insertion of dorsal fin |
| 5 | Edge of last hard spine |
| 6 | Insertion of soft rays |
| 7 | Maximum height of dorsal fin |
| 8 | Posterior insertion of dorsal fin |
| 9 | Upper limit of caudal fin |
| 10 | Hypural joint |
| 11 | Lower limit of caudal fin |
| 12 | Posterior insertion of anal fin |
| 13 | Maximum height of anal fin |
| 14 | Anterior insertion of anal fin |
| 15 | Insertion of pelvic fin |
| 16 | Lower occipital edge |

regardless of taxonomy and dried in paper envelopes until they were photographed for further analyses.

High resolution digital pictures of the left lateral view of the mimetic fish and model samples were taken over a black background using a Nikon D700 equipped with AF-S 60-mm immersive lens and a stand table with a reference scale of 1 cm for the fish and models. The left lateral view of the models was defined as the "dorsal view of leaves with the petiole oriented to the right." Artificial light was used to avoid shading morphological structures.

## Data analyses

Sixteen landmarks (LM) were established for the mimetic fish and models using ImageJ v. 1.47 software for geometric morphometrics purposes (*Abramoff, Magelhaes & Ram, 2004*). Homologous LM for the mimetic fish were marked obeying the morphological structures constrained or related to mimetic behaviour to cover the fish general outline profile, including peripheral structures (Fig. 1A and Table 1). We established equidistant 16 semilandmarks (SLM) for each model using the ImageJ grid tool to cover all lateral profiles of the model, obeying the same marking distribution as for the mimetic fishes (Fig. 1D). Raw coordinates LM and SLM data were implemented in MorphoJ v. 1.02n software (*Klingenberg, 2011*), where preliminary adjustments, such as the Procrustes fit, and creation of the data matrix, were done. The morphometric comparisons among the fish and models were not intended to analyse homologous patterns, as we were interested in shape similarities randomly shared among the mimetic fish and their respective models distributed in the same environment, from a geometric morphometrics perspective. Therefore, the necessity of marking peripheral anatomic structures in the mimetic fish,

instead of fins insertions only, in order to check for general appearance of mimetic fish with the plant models.

Data analyses were performed with Geomorph v. 2.0 software (*Adams & Otarola-Castillo*, *2013*). A post-hoc general Procrustes analysis (GPA) and principal components analysis (PCA) were run followed by analysis of variance (ANOVA) to compare the mimetic fish and models plotted together in the analyses. Also, a linear discriminant function was run, in order to visualize how close were these group associations, using the package MASS v. 7.3-42 (*Venables & Ripley*, *2002*).

In addition, individual TL and relative body area (BA, cm$^2$/TL) of the fish and models were calculated using ImageJ to establish interdependent comparisons among the fish species and plant debris via analysis of covariance (ANCOVA), followed by a linear discriminant analysis (LDA), to accurately predict whether the mimetic fish can be misclassified as a model. BA was chosen because of its importance for discriminating teleost aggregations (*Gómez-Laplaza & Gerlai*, *2013*). Fish were measured from the tip of the snout to the edge of the caudal fin (TL), and models were measured from edge to edge and considered TL. All statistical analyses were conducted in 'R' v. 3.1.3 (*R Development Core Team*, *2015*), and all relevant data for the current analysis are available within this paper (Dataset S1).

## RESULTS

Mimetic fish were observed mimicking plant debris near the water surface in all extensions of the port of Honmura. The mimetic assemblages resembled the models in shape, colour and drifting movements, having shared the same environment during the entire sampling period. All fish drifted among fallen plant debris near the water surface.

The visual GPA analysis indicated a significant variance in the shape configurations among the different models (Fig. 2A) and mimetic fish (Fig. 2B). All-pooled data showed a relative tendency of the mimetic fish to resemble plant debris with ∼24% of the variation explained in PC1 and ∼10% of the variation explained in PC2 (ANOVA $F_{2,52} = 40.97$, $P < 0.001$, Fig. 2C), yet remaining morphologically distinct, as observed in the GPA analyses.

BA of the mimetic fish and models regressed against TL revealed a significant interdependency (ANCOVA, $F_{2,96} = 92.06$, $P < 0.001$; Fig. 3), where juvenile *L. surinamensis*, *P. orbicularis* and *C. maculata* have shown a size gradient, sharing similar BA with round and elongated leaves of different sizes, accordingly to different growth stages of each mimetic fish species, with some deviation observed for the round leaf models. These results were corroborated by LDA, which has shown high similarities in shape of mimetic fish and models, with a 52.52% probability of misclassification among the observed individuals. Details on both ANCOVA and LDA can be found in Dataset S1.

## DISCUSSION

The present results show shape heterogeneity among mimetic fish and plant models, with a significant level of similarity shared in their general external shape profile. Such results are highly expected, as mimetic behaviour is more likely to be driven by a combination of factors (i.e., shape, colour and movements) than solely by morphological attributes

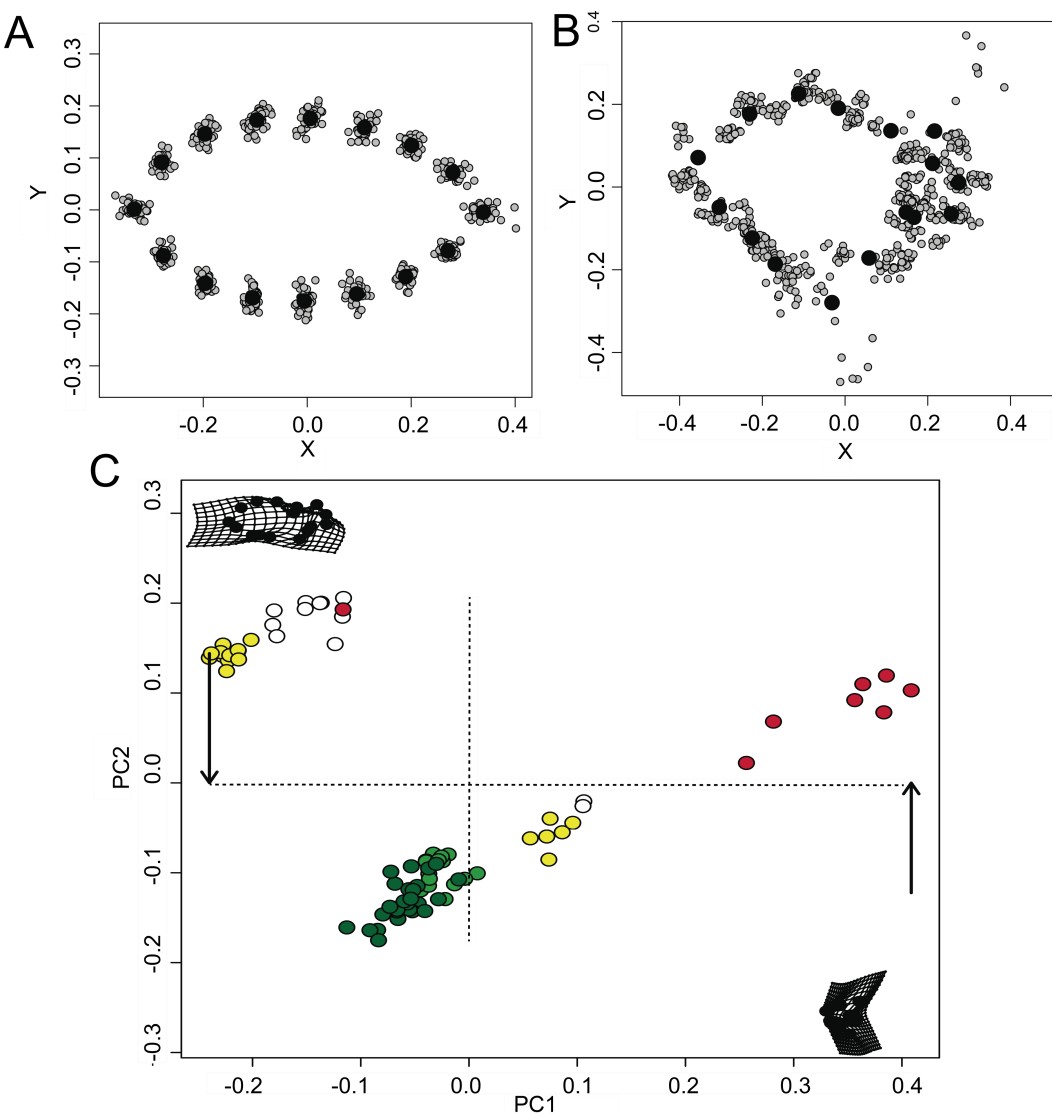

**Figure 2** **Morphometric relationships among mimetic fish and plant models.** Diversity of shapes observed for the models (i.e., floating plant debris) (A) and fish mimics (B), via a general Procrustes analysis (GPA); and principal components analysis (PCA; (C)), of all- pooled data indicating a high tendency for shape similarities shared by the mimetic fish and models (i.e., floating plant debris), where green plots represent leaf models (dark green representing rounded leaf models and lighter green representing elongated leaf models). Mimetic fish are represented by *Lobotes surinamensis* (yellow), *Platax orbicularis* (red), and *Canthidermis maculata* (white).

(*Wickler*, *1968*; *Pasteur*, *1982*). Although the importance of floating plant debris for passive transportation, providing shelter and feeding grounds for fish in coastal environments has been evaluated (*Castro, Santiago & Santana-Ortega*, *2001*; *Vandendriessche et al.*, *2007*), the closeness of these interactions has not been investigated, particularly regarding plant resemblance by fish.

Arching effects due to fixation protocols are known to strongly influence geometric morphometric analyses (*Valentin et al.*, *2008*). Although we have combined data from

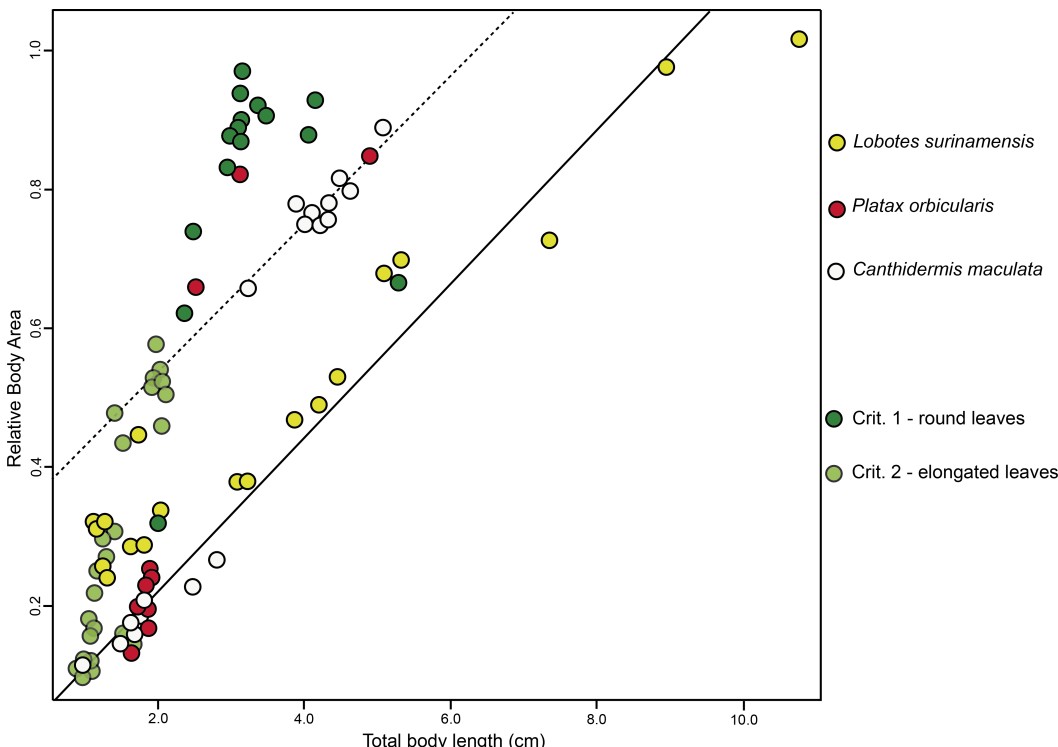

**Figure 3  ANCOVA.** Similar relative body area values were observed among the models (i.e., floating plant debris) and mimetic fish, where mimetic fish are represented by *Lobotes surinamensis* (yellow), *Canthidermis maculata* (white) and *Platax orbicularis* (red), and plant models are represented by green plots (dark green representing rounded leaf models and lighter green representing elongated leaf models).

museum specimens with our own samples, we have selected only intact individuals for the present analyses. According to observed shape similarities shared among the mimetic fish and models, it was clear that the present fish assemblage accompanied their respective models, being probably dependent on drifting plant material for survival, also suggested by the linear model of covariance shared amongst drifting fish and plants. While not the primary goal of the present study, such association might suggest an allometric dependence for the plant mimetic species, at least until a given ontogenetic stage when such fish species suffer significant changes in morphology and behaviour, cessing with the mimetic association with plants (*Barros et al.*, *2015*).

The concepts regarding mimetic behaviour are still a matter of discussion, as it is difficult to define a case of mimetic association using only a shape resemblance to another animal/inanimate object (*Skelhorn, Rowland & Ruxton*, *2010*; *Skelhorn et al.*, *2010*), especially in marine systems (*Robertson*, *2013*; *Robertson*, *2015*). The observed species herein not only presented good shape similarity with the models, but also behaved alike, via drifting movements along with their respective models, far away from being ''inanimate'' (B Barros, pers. obs., 2004–2006; Video S1). Close resemblance of fish to their models in shape and drifting behaviour at the water surface environment could confuse visually oriented predators through the camouflage effect. Thus, ''mimetic behaviour'' was a valid classification in the present case.

All species tested in the present study, such as *L. surinamensis* (Lobotidae), *C. maculata* (Balistidae) and *P. orbicularis* (Ephippidae) have been described previously as resembling dried leaves in shallow water (*Uchida*, *1951*; *Breder*, *1946*; *Randall & Randall*, *1960*; *Barros et al.*, *2008*; *Barros et al.*, *2011*; *Barros et al.*, *2012*), and are commonly found in the surveyed area (*Motomura et al.*, *2010*).

Although coastal fish resembling a plant via cryptic colouration has been an intriguing subject since the early reports, the present study is the first attempt to establish analytical comparisons between mimetic fish and models at the morphometrics level. *Kelley & Merilaita* (*2015*) suggested that successful crypsis in fish is more likely achieved through colouration, via a background matching effect. Although we did not test the predation rate of mimetic fish nor for any colour influence, our results add relevant information, in which background matching is achieved not only by cryptic colouration (*Breder*, *1946*; *Randall & Randall*, *1960*; *Randall*, *2005b*), but also through shape and behavioural resemblance of mimetic fish to their respective models. The present level of protective camouflage shared by the fish assemblage analysed herein might be important against potential aerial and bottom predators, as background colour matches surrounding environments (*Donnelly & Whoriskey Jr*, *1991*; *Cortesi et al.*, *2015*; *Kelley & Merilaita*, *2015*). However, no predatory attempt by a bird species has been observed. Further experiments and field observations of all observed species are necessary to test this assumption.

The co-occurring mimetic assemblages observed herein are a typical example of convergent evolution in a coastal environment (*Endler*, *1981*; *Hamner*, *1995*; *Johnsen*, *2014*). Some taxa analysed undergo numerous morphological and ethological changes. For example, *P. orbicularis* adults inhabit deeper environments, changing in both shape and behaviour within the settlement (*Kuiter & Debelius*, *2001*; *Barros et al.*, *2011*). As major morphological changes are usually expected through ontogeny of several fish groups (*Galis*, *1990*; *Loy et al.*, *1998*; *Comabella, Hurtado & García-Galano*, *2010*; *Leis et al.*, *2013*; *Nikolioudakis, Koumoundouros & Somarakis*, *2014*; *Barros et al.*, *2015*), resemblance to leaves by the fish species observed here may be crucial for first settlement, as it could improve survival chances (*Johnsen*, *2014*).

The Kuroshio Current is regarded as a key factor for passive transportation of masses of plant and algae material and juvenile fishes closely associated with, as such ichthyofauna use the plant debris as both shelter and food source (*Kimura et al.*, *1998*). Strictly morphological studies are ineffective for providing all of the clues necessary to interpret the natural history of most living organisms (*Scholtz*, *2010*). The present observations support fundamental information on the distributions of these fish species during early stages, their life history and evolutionary paths if combined with mimetic fish and model ethological and ecological data that are available for some taxa (*Barros et al.*, *2008*; *Barros et al.*, *2011*, *Barros et al.*, *2012*). Although refinements to the methodologies are necessary, this new comparative approach may stimulate discussion of morphology as a predictor of ecology (*Douglas & Matthews*, *1992*; *Gibran*, *2010*; *Oliveira et al.*, *2010*).

## ACKNOWLEDGEMENTS

We thank all members of the Kuchierabu-jima Island community, particularly M Yamaguchi, the crew of the Laboratório Multi-Imagem and FRR de Oliveira (UFPA), and A Akama (MPEG) for criticism and logistic and technical support during this study. We are deeply grateful to Dr. Hiroyuki Motomura (Kagoshima University Museum) for supplying additional samples, and Dr. Yuki Kimura and Ms. Misaki Fujisawa (Hiroshima University) for technical support. This study is in memory of Dr. Kenji Gushima.

### Funding

This study was financially supported by CAPES (process #6718-10-8), FAPESPA (process # 456780/2012), and the following research projects: "Fluxos (Água, Sedimentos, Nutrientes e Plâncton) Amazônicos ao longo do Continuum Rio-Estuário-Costa e Implicações para a Biodiversidade Vegetal Costeira Amazônica" (Programa CAPES Pró-Amazônia: Biodiversidade e Sustentabilidade—Edital 047/2012) AUXPE no. 3290/2013, and "Descoberta de um Novo Bioma Marinho Amazônico" (Programa IODP/CAPES-Brasil—Edital 038/2014) Processo no. 88887.091707/2014-01. The funders had no role in study design, data collection and analysis, decision to publish, or preparation of the manuscript.

### Grant Disclosures

The following grant information was disclosed by the authors:
CAPES: process #6718-10-8.
FAPESPA: process #456780/2012.

### Competing Interests

The authors declare there are no competing interests.

### Author Contributions

- Alexya Cunha de Queiroz conceived and designed the experiments, performed the experiments, analyzed the data, contributed reagents/materials/analysis tools, wrote the paper, prepared figures and/or tables.
- Yoichi Sakai conceived and designed the experiments, analyzed the data, contributed reagents/materials/analysis tools, wrote the paper, reviewed drafts of the paper, sampling.
- Marcelo Vallinoto conceived and designed the experiments, analyzed the data, contributed reagents/materials/analysis tools, wrote the paper, reviewed drafts of the paper.
- Breno Barros conceived and designed the experiments, performed the experiments, analyzed the data, contributed reagents/materials/analysis tools, wrote the paper, prepared figures and/or tables, reviewed drafts of the paper, sampling.

## Animal Ethics

The following information was supplied relating to ethical approvals (i.e., approving body and any reference numbers):

As there is no national Japanese licensing framework, samples were collected following the "Guidelines for Proper Conduct of Animal Experiments" set out by the Hiroshima University Animal Research Committee, which are based on international ethical standards (*Jenkins et al.*, *2014*), and only after obtaining local community permission.

## Data Availability

The raw data and video has been supplied as Supplementary Files.

## Supplemental Information

Supplemental information for this article can be found online at http://dx.doi.org/10.7717/peerj.2268#supplemental-information.

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
