# Peer review of "Morphometric comparisons of plant-mimetic juvenile fish associated with plant debris observed in the coastal subtropical waters around Kuchierabu-jima Island, southern Japan"

_PeerJ, doi:10.7717/peerj.2268_

## Round 0.1 · original submission · Major Revisions

Three reviewers have now commented on your manuscript. While reviewer 2 is positive, reviewer 3 feels you need to better explain your experimental design, as well as parts of the result and discussion. Most importantly, reviewer 1 mentions serious issues with the number of specimens examined and the statistics used to examine your hypothesis and judge significance.

Despite such disparate comments, most of the issues mentioned by the reviewers stem from the low numbers of specimens and analyses. I agree with reviewer 3 that it is likely hard for you to supplement your specimen numbers, but I also agree with reviewer 1 that your small numbers of specimens and large number of factors may obscure your results, and the statistical methods utilized are not the most appropriate.

Thus, I suggest the following actions:

1. Is it possible to supplement your data with specimens from museums? Japan has numerous museums and many specimens and much data are potentially available. More specimens would be the easiest way to alleviate most of the issues of this manuscript.
2. If 1 is not possible, then please re-examine your statistical methods used. I would like to see either different methodologies employed, or else an explanation (in the M&M) of why the current statistical methods can be applied in this situation.
3. As well, please include an open and frank discussion of the small numbers of specimens and problems with your data. Finally, I agree with reviewer number 1 that your current discussion far overstates your findings and is too ambitious given your low specimen numbers. Please thoroughly and critically re-examine this part of your manuscript in particular. Please note that failure to address numbers 1 to 3 in a thorough fashion may well result in rejection after the next round of reviews.

Other comments made by all reviewers seem helpful and fair, and will also help you revise your paper.

For the above reasons, my decision is "major revision".

Reviewer 1 ·

Basic reporting

References: The paper is too long and there are too much references that difficult the reading of the manuscript.

The discussion is too long and repetitive in some points. I think that it can be shortened about a 25 % without losing information.
Discussion is to ambitious, more if you are taking into account that only 15 specimens from 3 species were sampled, one of which with only two individuals.

Experimental design

My major concern refers to the small sample size, only 15 individuals belonging to 3 fish species were captured in one week field sampling. One of the species is only represented by two individuals. At this point is difficult to justify the paper and the conclusions assumed.

How is the natural shape variability affecting results? It is not possible to measure natural variability.

Due to the small sample size but the large number of landmarks the results are completely overestimated.

Additionally, no information about the sampling is provided, are for instance fish and mimetic plants captured in the same samples?

Statistical methods are not the most appropriate for reduced sample sizes.

Validity of the findings

My major concern refers to the small sample size, only 15 individuals belonging to 3 fish species were captured in one week field sampling. One of the species is only represented by two individuals. At this point is difficult to justify the paper and the conclusions assumed.
How is the natural shape variability affecting results? Can you measure variability with only 2 fish species?
Due to the small sample size but the large number of landmarks the results are completely overestimated.
Additionally, no information about the sampling is provided, are for instance fish and mimetic plants captured in the same samples?
Statistical methods are not the most appropriate for reduced sample sizes.

Additional comments

Interesting MS dealing with fish morphometrics. However, some majors concerns arisen after my reading of the manuscript:
- Methods and Data analysis. My major concern refers to the small sample size, only 15 individuals belonging to 3 fish species were captured in one week field sampling. One of the species is only represented by two individuals. At this point is difficult to justify the paper and the conclusions assumed.
How is the natural shape variability affecting results? Can you measure variability with only 2 fish species?
Due to the small sample size but the large number of landmarks the results are completely overestimated.
Additionally, no information about the sampling is provided, are for instance fish and mimetic plants captured in the same samples?
Statistical methods are not the most appropriate for reduced sample sizes.

- References: The paper is too long and there are too much references that difficult the reading of the manuscript.

Specific Comments:
Abstract
Lines 24-25: Change “Lobotes surinamensis (Lobotidae), Platax orbicularis (Ephippidae) and Canthidermis maculata (Balistidae), three plant-mimetic fish species, were compared” to “Three plant-mimetic fish species, Lobotes surinamensis (Lobotidae), Platax orbicularis (Ephippidae) and Canthidermis maculata (Balistidae) , were compared”.
Line 27: Change “plant parts” to “plant debris”.
Line 31: by a linear model?

Keywords
Avoid using the same words as in title, thus remove Morphometrics.

Introduction
Here and trough the manuscript. Remove some references, for instance in lines 47-48, 5 references are provided, and in lines 54-55, nine references are provided. The amount of references is unnecessary, dificulting the reading.
Introduction needs to be reworded and maybe shorthened, for instance lines 48-50 (“Therefore, many…”) can be deleted without losing information.
Lines 68-104: Can be shortened and summarized (up to ca. 33 %) without losing information.

Methods
Lines 113 and through the text: add an space between number and %.
Lines 124-126: Only 15 fish individuals were captured during one week sampling. This is a major issue (see general comments).
Lines 124-125: What is the number between brackets? SD or SE? Please include.
Line 127 and through the text: Change “n” to “n” in cursive.
Lne 133: Why left and not right when photos were taken?
Line 141: Sixteen landmarks but only 6, 7 or 2 specimens… the ration is completely overestimated.
Line 158: Data analyses methods are not the most appropriate to your small sample size
Line 164: Why are you using TL in statistical analyses, but relative body are is standardized by SL?

Results
See previous comments.
Additionally I recommend joining results to discussions in a results and discussion section.

Discussion
The discussion is too long and repetitive in some points. I think that it can be shortened about a 25 % without losing information.
Discussion is to ambitious, more if you are taking into account that only 15 specimens from 3 species were sampled, one of which with only two individuals.

·

Basic reporting

First, I would like to congratulate the authors on an extremely well prepared manuscript.
The abstract is well written and provides a good window to the paper. I note that the authors use plant-mimetic, mimicry, mimesis and mimetic fish for cases that some purists would sort among protective resemblance given a strict definition of fish mimicry as only occurring when a fish evolves to resemble another. However, I agree with the way the authors have dealt with this issue and for the sake of readability I think the terms chosen are useful.
The introduction is well written and gives sufficient background for the study. The references chosen are appropriate and different views are well represented.
Results: Figures are relevant and the quality is high. I would have liked to see the nice study site figure provided in the supplements included in the main paper!
Line 170: repeat what the LM abbreviation means to keep the reader on track.

Raw data is supplied.

Experimental design

The hypotheses are well defined and within reach in terms of what the study can deliver.
The research is original and useful as an example of taking what subjectively meets the eye into a quantitative framework and thus seek to increase the objective validity of a science deemed largely anecdotal by some.
Materials and methods are described with sufficient detail and information. Analyses seem appropriate and are clearly and credibly presented.

Validity of the findings

The discussion is interesting and well written.
Line 194-198: This sentence is useful and true but could be rewritten to state simply that these are cases where the plant-mimetic species outgrow their plant debris model and thus the mimesis ceases. There are plenty of examples where fish are only mimics during vulnerable life stages, and reference to juvenile fish mimicry could fit in here.
Line 202: Roberston, a typo.
Line 204-205: personal observation of behaviour, is there video material available? A link to a video clip made available online would have been nice if it exists, to support the case for “mimetic behaviour” (line 207). Cases where mimetic fish modify their behaviour in order to increase the resemblance to a model are well known and could be cited to lend further support.
Data is robust, statistically sound and controlled.

Additional comments

My suggestions above are for consideration only and whether they are used in a minor revision or not should by no means preclude acceptance of the paper which in my opinion may be published forthwith.

Reviewer 3 ·

Basic reporting

This manuscript focused the relationships between plant-mimetic fish species and plant parts, and was interesting philosophy for the fish morphological work on the bases of statistic methods.
However, some of the parts on the manuscript are difficult to understand, and need more careful explanations, especially in the results and discussion.

Experimental design

The selection of the species is bit difficult, the number of C. maculata are fewer than other two species, and may making problems on the statistic methods.
However, I can understand to collect a number of this species are so difficult, so is nonsense to order to add more data for this species. Nonetheless, this manuscript seem diverting to touch this species especially in results. The authors should carefully explain "what can read from figure" in results, and express their opinion of why the results differed (or not differed) in discussion.

Validity of the findings

Of cause, the facts these three fish species are mimicking plant parts, are widely known in many publications. However, the interest of the manuscript is to show the fact in science.

Additional comments

The maps and photos (Fig. 1) needs to modify. This figure may good for power point presentation, but no good for manuscript.
For examples, some bays of the map are still white colored; border of photo A and B is unclear; the locality of Kuchierabu-jima is impossible to understand from this figure.
Other comments are written in the manuscript.

Annotated reviews are not available for download in order to protect the identity of reviewers who chose to remain anonymous.

---

## Round 0.2 · Major Revisions

I have received comments back from one reviewer who had reviewed the previous version of this manuscript. The reviewer has noted you have made much progress addressing the concern on the number of specimens, but there are still some outstanding issues remaining.

In particular, the reviewer makes an excellent comment on the preservation issues surrounding the use of specimens from different collection events, and possibly under different protocols or fixatives. At a bare minimum, you must address how such issues could influence your results, and temper your discussion accordingly. I realize you are in somewhat of a 'no-win' situation, as the sample numbers from the first submission resulted in you adding these new specimens from other collections. Still, as reviewer 1 has correctly pointed out, such issues could directly and strongly influence your results, and they must be addressed.

Reviewer 1 ·

Basic reporting

The authors have done a strong effort to fix all isues raised during the reviewing process, however some points remain unclear (see below).

Experimental design

The authors have done a strong effort to fix all isues raised during the reviewing process, however some points remain unclear (see below).

Validity of the findings

I am still concern about the sample size and sampling procedures. There are a lot of papers showing that animal fixation protocols can modify and alter morphological structures (e.g. ethanol, formaldehyde etc…). Authors explain that some of the samples are coming from museums or other samplings, but nothing its said about animal preservation. Was all the samples preserved following the same protocol? If not, how different protocols can affect results?
After checking Figure 2 (ANCOVA), there is something missed. Red and both green groups don’t follow lines plotted, and have a greater slope.

I am still concerned about the proportion between landmarks and sample size. Sixteen landmarks and 20 Lobotes surinamensis, 18 Platax orbicularis, 12 Canthidermis maculate.

I a followed the procedures you get 15 sampled individuals, 24 from Kagoshima University Museum and 11 from previous surveys. The total sampled fish is 50 not 51 as present in the paper. According to additional data there are 12 additional individuals for Platax orbicularis, not 11 as indicated in Methods section.

---

## Round 0.3 · Minor Revisions

Your manuscript is almost ready to be published, and there are just a very few small issues to address. I have listed these below.

1. I have heard back from one reviewer, who now finds your paper acceptable for publication except for the ANCOVA figure.

In a previous round of reviews, this reviewer mentioned:
"After checking Figure 2 (ANCOVA), there is something missed. Red and both green groups don’t follow lines plotted, and have a greater slope."
Your response indicated that the lines were generated by the software utilized, and are correct.
I wish to confirm with you that this is the case, as the reviewer has again mentioned ANCOVA figure in their review. If you can confirm this for me, then I will leave the figure as is and this issue has been dealt with.

2. Regarding the issue of different sample preservation among different specimens, you must at least address this with one or two sentences as a "caveat emptor". I have marked the PDF with a possible suitable location, please check the attached file.

3. There are some small English edits to be done - these are also visible in the attached PDF file.

I look forward to viewing the revised version.

Reviewer 1 ·

Basic reporting

All comments have been addressed.

Experimental design

All comments have been addressed, except those on the ANCOVA analysis.

Validity of the findings

All comments have been addressed.

Additional comments

All comments have been addressed, except those on the ANCOVA analysis.

---

## Round 0.4 · accepted · Accept

I look forward to seeing this manuscript in published form.